# A Comparative Study on the Quality of Life of Survivors of Stroke and Acute Myocardial Infarction

**DOI:** 10.3390/healthcare12020254

**Published:** 2024-01-19

**Authors:** Eva Lourenço, Mário Rui dos Mártires Sampaio, José Luis Sánchez-Ramos, Emília Isabel Martins Teixeira da Costa

**Affiliations:** 1Intensive Care Unit, Algarve University Hospital Center (CHUA), 8000 Faro, Portugal; 2Faculty of Medicine and Biomedical Sciences, Algarve University, 8000 Faro, Portugal; 3Tavira Health Center, Algarve Health Administration, 8800 Tavira, Portugal; msampaio@arsalgarve.min-saude.pt; 4Nursing Department, Huelva University, 21001 Huelva, Spain; jsanchez@uhu.es; 5Nursing Department, Algarve University, 8000 Faro, Portugal; eicosta@ualg.pt; 6Health Sciences Research Unit: Nursing, 3000 Coimbra, Portugal

**Keywords:** Quality of Life, acute myocardial infarction, stroke

## Abstract

Background: Worldwide, cardiovascular diseases, particularly acute myocardial infarction and strokes, lead to significant fatalities. Survivors often experience profound impacts on various aspects of their lives, making the assessment of their Quality of Life crucial for understanding their condition and adaptation to the illness. Methods: A community-based, descriptive cross-sectional study was conducted to compare how survivors of stroke and acute myocardial infarction perceive their Quality of Life. The Portuguese version of the World Health Organization Quality of life instrument was administered to 204 acute myocardial infarction and stroke survivors. Clinical and sociodemographic variables were also compared. Results: Statistically significant differences (*p* < 0.05) were noted in overall, physical, and psychological aspects of Quality of Life between the two groups, with stroke survivors consistently showing lower values in these dimensions. Conclusions: The Quality-of-Life perception of stroke and acute myocardial infarction survivors may be affected by several sociodemographic and clinical factors, and the results support the idea that the vascular event conditions the person’s Quality of Life, particularly in variables related to the person’s physical condition, functionality, and autonomy.

## 1. Introduction

The global occurrence of an aging population [1] triggers notable changes in healthcare necessities. The altered population and health landscape causes more cases of long-term health issues. These conditions have a big impact and are reshaping healthcare systems globally [2]. At the same time, there is a concerning surge in specific vascular risk factors, like unhealthy eating patterns and obesity, significantly influencing the overall health of the populace. Indeed, we are facing a continuous increase in cardiovascular diseases, stemming from multifaceted and intricate causes that intersect various societal sectors. Similarly, its impact manifests across multiple levels, reflecting the complex nature of this issue [3,4,5].

Despite this, there seems to be a decreasing trend in mortality from this type of health problem, because of greater dissemination and adherence to preventive measures and the organization of health services with the creation of stroke and acute myocardial infarction (AMI) codes, percutaneous coronary intervention, and stroke units [6]. However, these two diseases are still responsible for high morbidity and mortality, particularly in industrialized countries, being the leading cause of death in both sexes in our country, Portugal [1].

Stroke is defined as an episode of neurological dysfunction, caused by cerebral, spinal, or retinal infarction [7]. It is currently one of the leading causes of death and physical and cognitive disability. Many patients require rehabilitation, and around 70% are unable to return to work [1]. Quality of life (QoL) in post-stroke depends on the severity of neurological deficits, but it is usually low and although it may improve with the resolution of sequelae, it may never reach the pre-event level [8]. The impact on mental health is reflected by the incidence of depression and anxiety, and changes in professional activity and social roles are among the main complications that occur in the post-stroke period and that negatively impact the individual’s well-being. For a large proportion of stroke survivors, starting a new life with the dysfunctionalities and disabilities generated by the disease can have a huge impact on their perceived Quality of Life [9]. The assessment of this variable has been identified as an important index of post-stroke outcomes and proves to be more important for the individual than any other parameter [10]. 

On the other hand, in the group of cardiovascular diseases, acute myocardial infarction (AMI) is the myocardial necrosis secondary to prolonged ischemia [11]. Despite the improved management of ischemic heart disease in recent years, it also remains a major cause of death with significant personal, family, social, and economic repercussions. The survivors of AMI also present with a decrease in the perception of their QoL in relation to the physical and psychological limitations inherent to the deterioration of cardiac function [12]. The degree of progression of coronary artery disease appears to be related to a decrease in perceived QoL, probably because more advanced coronary atherosclerotic disease leads to more cardiac semiology [13]. It is important to highlight that in post-infarction, by re-establishing coronary flow, surgical myocardial revascularization reduces or eliminates anginal symptoms, providing greater functional capacity and improved QoL, which may even be better after the event [14].

In this scenario, it becomes evident that healthcare providers need to understand how these patients perceive their QoL. This understanding is crucial for devising effective strategies aimed at enhancing health and reducing the adverse effects that these diseases impose on the lives of patients, their families, and the community [15]. Enhancing QoL stands as a focal point in current health policies, encompassing both illness prevention and health promotion. Its assessment serves as a metric to gauge the effectiveness of treatments and disease progression, offering deeper insights into a patient’s adaptation and understanding of their condition [16]. By employing tools that evaluate QoL, it becomes feasible to track the effectiveness of selected interventions based on individual perceptions. Additionally, these tools enable a comparison of treatment benefits, simplifying the assessment of their effectiveness across different therapies [17].

The purpose of this study was to analyze and contrast how individuals who have survived stroke and acute myocardial infarction perceive their QoL.

## 2. Materials and Methods

### 2.1. Study Design, Setting, and Participants

From February to September 2018, a community-based, descriptive cross-sectional study was conducted involving stroke and AMI survivors. The study encompassed individuals from a public primary health care unit located in the southern Algarve region. This center caters to approximately 26,100 residents [18]. In Portugal, primary health care is universally accessible without charges and serves as the primary setting for regular patient monitoring across various acute and chronic health conditions.

After the due formal authorizations, data were collected. The institutional electronic database was used to identify the patients meeting the following inclusion criteria: >18 years at the time of diagnosis; stroke diagnosed at least 3 months apart or acute myocardial infarction at least 6 months apart; followed at the public primary health care unit; voluntarily accepting to participate in the study upon adherence to the Informed Consent Form and answering more than 80% of the questions. The criteria for including stroke survivors followed the Stroke Recovery and Rehabilitation Roundtable task force’s recommendations [19], specifying that beyond 3 months post-event, individuals typically reach a plateau phase in their recovery. For AMI survivors, the inclusion criteria aligned with the guidelines from the Portuguese Society of Cardiology [20], considering the long-term phase in cardiac recovery to begin after 6 months.

The exclusion criteria for both groups were cognitive deficits, an altered state of consciousness or inability to establish effective communication, and incomplete records. The researchers chose not to establish exclusion criteria tied to the duration following the event, prioritizing inclusivity to embrace a spectrum of diverse experiences. Vascular events manifest uniquely in individuals, leading to distinct paths of recovery. By avoiding a defined time limit post-event as an exclusion standard, the study aimed to encompass a wider spectrum of experiences, acknowledging different stages of recovery. Additionally, excluding individuals solely based on the time elapsed since a vascular event might be seen as unjust or biased. Welcoming participants across varied post-event durations aligns with the ethical considerations of inclusivity in research.

After identification of the patients who met the inclusion criteria for the study, they were contacted personally, during the medical consultation or by telephone, and asked if they agreed to participate in this study. If they were in accord, an informed consent form was signed, and questionnaires were completed. When the primary point of contact was through telephone communication, we endeavored to reach each individual on up to three occasions. Those individuals for whom contact could not be established after these attempts were then classified as unreachable, resulting in a total count of 23 individuals falling into this category.

Over a period of about seven months, 103 questionnaires were obtained from stroke survivors, with 101 questionnaires from AMI survivors. These questionnaires were administered during nursing or medical consultations. Some patients, as we mentioned before, were also contacted by telephone and came to the health center to answer the questionnaire. In the case of patients who were institutionalized and whose degree of dependency made it difficult for them to travel to the health center, the research team went to the institutions to collect the data. 

Considering the academic setting of this research, collecting data encountered limitations in resources, specifically time and funding. This is a significant factor behind the decision not to prolong the data-collection period. Moreover, after over six months of data collection, we reached a point where we had maximized respondent intake given the prevailing circumstances. Consequently, this prompted the conclusion of this phase in the research process.

### 2.2. Data Measurement

Three questionnaires were used as assessment instruments: an ad hoc socio-demographic questionnaire (completed by the patient and/or researcher), clinical questionnaire (completed by the researcher), and World Health Organization Quality of Life-brief scale–WHOQol-bref (completed by the patient with or without the help of the researcher or by the researcher with the help of the patient).

The collection of socio-demographic variables involved elements of the research team and the patient. Some of these variables are available in the National Health System electronic database and were collected by the researcher. Those that were not in electronic format were provided by the patient or family. The variables collected included age, gender, marital status, formal education, employment status, income, and existence of a caregiver.

The variables related to the clinical questionnaire were collected from the electronic database. Most of the clinical variables are similar in the two groups of patients, except for the following variables: neurological syndrome, sequelae and current mRankin, which are intended for stroke survivors, and the variable physical limitation according to the NYHA Functional Scale of Heart Failure, which is applied to the other group of patients. The modified Rankin scale (mRanking) is a scale commonly used to measure the degree of dependence in activities of daily living in people who have suffered a stroke or other causes of neurological disability. It is also used by hospital systems to assess patient rehabilitation needs. This scale ranges from 0 to 6, from symptom-free to death [1].

Among the different clinical variables described, we highlight the paramount importance of the “Via Verde” program for stroke and AMI cases. This initiative serves as a vital guarantee in our country, ensuring that individuals exhibiting signs and symptoms of these vascular events receive precise care and are swiftly directed to the most appropriate hospital unit for their specific needs. As per the Portuguese Ministry of Health [21], the accurate referral of stroke and AMI cases via the National Institute of Medical Emergency (INEM) results in crucial time savings for optimal therapeutic outcomes. This essence encapsulates the Via Verde concept and substantiates the extensive organizational endeavors dedicated to its success. The QoL of survivors of ictus and AMI was assessed using the WHOQoL-bref [22]. Initially, the World Health Organization developed the WHOQoL-100 instrument, which is constituted by six dimensions (physical, psychological, level of independence, social relationships, environment, and spirituality), which branch into 25 facets, each with four questions totaling a questionnaire with 100 questions. This first instrument proved to be too long, and the shortened WHOQoL-bref version [23] emerged to assess QoL in a more accessible way, over a shorter period of time and while maintaining satisfactory psychometric characteristics.

Thus, the shortened instrument is a research tool in QoL with multidimensional and cross-sectional applicability, in terms of the respondent type. The WHOQOL-bref is organized in facets, which include 26 questions, of which 24 are organized in the physical, psychological, social, and environmental dimensions, and the remaining two items are part of the general facet, which provides for the assessment of the general QoL and general perception of health. The response is given under the guidance of a Likert-type scale of intensity from 1 to 5. Once all responses are obtained, the score for each domain is summed using the user manual and the syntax of the scale [22].

The results for the dimensions show the perception of QoL on an individual basis, directed at the domain under analysis. Higher scores indicate a better perception of QoL, once the dimensions are positively arranged. The questions that are formulated inversely were previously recoded. The WHOQoL-bref allows for three data collection options: self-administered, interviewer-assisted, and interviewer-administered. All forms were used by the researchers for data collection in the present study.

In the original instrument, the reliability coefficients for the different dimensions ranged from 0.68 to 0.82 [23]. In the study that validated the WHOQoL-bref for the Portuguese population [24], Cronbach’s alpha values ranged from 0.64 to 0.87. In the present study, we obtained a Cronbach’s alpha ranging from 0.68 to 0.89 (general, α = 0.862; physical, α = 0.882; psychological, α = 0.892; social, α = 0.680; environmental, α = 0.830). To meet the defined objective, with the analysis of the relationship between the variables, data analysis was carried out using a descriptive and inferential analysis. In the descriptive analysis, absolute frequencies (n) and percentages (%), as well as measures of central tendency (averages and the minimum and maximum limits, when relevant), are presented. The statistical analysis of the data obtained was processed using the software SPSS (Statistical Package for the Social Sciences) version 25.

## 3. Results

### 3.1. Sociodemographic and Clinical Variables

Out of the 224 individuals who met the inclusion criteria and were reachable, 204 responded to the survey. Of these, 103 (50.5%) were stroke survivors and 101 (49.5%) were AMI survivors. The average age of the study participants was 68.7 years (SD = 12.1), with a minimum of 38 years and a maximum of 94 years of age. The average age at the event was 66.1 years (SD = 11.9), with a minimum of 34 years and a maximum of 92 years of age. It is noted that the youngest patient with a vascular event, aged 34 years, had a hemorrhagic stroke. When it comes to the time passed since the cardiovascular event, our findings revealed that stroke survivors, on average, experienced it about 29.42 months before (SD = 31.806), ranging from a minimum of 3 months (meeting our inclusion criteria) to a maximum of 144 months. Conversely, for AMI survivors, the average duration since the event was approximately 31.90 months (SD = 37.316), ranging from a minimum of 6 months (meeting our inclusion criteria) to a maximum of 240 months.

Most of the sample was made up of men (69.6%). In both diseases, most respondents were married or of an unmarried couple (60.8%). 

Regarding the education attainment, the majority had up to six years of schooling (47.5%), and 36 patients (17.7%) had no education. Only 7.4% of patients attended university education. Most of the participants in the sample were retired (59.3%), consistent with the elevated average age. Of those surveyed, only 27.5% were professionally active, with a higher percentage of AMI survivors (38.6%) compared to stroke survivors (16.5%). Of all patients, 43.2% received a salary below the minimum wage (€580). Only 14.2% of the patients had a monthly income equal to or higher than three minimum wages. In terms of place of residence, 57.4% of patients lived in the city.

Considering the family unit, only 18.6% of the participants lived alone, with this percentage being higher for AMI survivors (25.7%). Twenty-two patients (10.8%) were in institutions due to lack of social support or physical and/or psychological dependency. Of the individuals surveyed, 35.3% reported having a caregiver, with this percentage being more relevant in stroke survivors with 49.5%. Table 1 illustrates the distribution of socio-demographic variables based on the type of cardiovascular event, highlighting the statistical differences identified among the diverse categories of each variable subsequent to conducting the chi-squared test.

We will now present the clinical variables according to the different cardiovascular events. We underline that several parameters were evaluated for the two groups, but the current mRankin applies only for stroke survivors and the physical limitation according to NYHA applies to AMI survivors. 

To assess whether AMI survivors were experiencing the semiology of heart failure, the New York Heart Association (NYHA) Heart Failure Functional Assessment Scale was used [25]. This scale is commonly employed as a method for the functional classification of patients with heart failure. It designates four functional classes (I, II, III, and IV), ranging from no limitation to inability to perform any physical activity without discomfort.

Concerning the clinical variables, we can start by noting that most patients (73.5%) were not admitted to the hospital for “Via Verde” (green pathway) activation. Even so, it is worth noting that the highest percentage of referrals was for AMI survivors (37.6%) compared to stroke survivors.

We found that, as expected, 98.0% of the participants had cardiovascular risk factors at the time of the vascular event. Only four (2%) patients had a vascular event without at least one associated risk factor identified in the initial examinations. The mean number of cardiovascular risk factors was 3.34, with a minimum of 0 and a maximum of 8. Three cardiovascular risk factors were found in 30.4% of patients, followed by four in 25% of patients. The most prevalent risk factors at the time of the vascular event were hypertension (75.5%) and dyslipidemia (64.7%), which, in fact, are known to be the most prevalent cardiovascular risk factors in the general population. They were followed by a sedentary lifestyle in 38.2% of patients, type 2 DM in 31.9% of patients, and smoking in 29.4%.

When discussing mental health as a crucial aspect of the QoL, we observed within our sample that over 25% of the participants possessed a documented medical history of a mental health condition, specifically depression and anxiety. Notably, survivors of stroke, exhibited markedly higher values in both depression and anxiety compared to the other group in the study.

With respect to post-event evolution, we can assert that slightly over half of the patients (52.5%) had a negative evolution with sequelae, with this percentage being more relevant in stroke survivors (70.8%). More precisely, when we evaluated the current mRanking, we see that 45.6% of stroke patients had at least some deficits that prevented them from carrying out the same activities as before the ictus. On the other hand, most of the patients who suffered from AMI (59.4%) did not show signs of heart failure in their daily activities according to the NYHA functional classification. Table 2 shows how clinical variables were distributed concerning the type of cardiovascular event, emphasizing the statistical differences observed among the different categories of each variable following the chi-squared test application. 

### 3.2. Comparative Analysis of the Perception of Quality of Life between Survivors of Ictus and AMI

After testing the normality of the distribution using the Kolmogorov–Smirnov test (*p* < 0.001), the differences in QoL perception between the two groups of cardiovascular event survivors were assessed with the Mann–Whitney U test, indicating statistically significant differences (*p* < 0.05) in the overall, physical, and psychological dimensions. Survivors of AMI displayed higher means in these areas, suggesting a better perception of their QoL compared to stroke survivors (Table 3).

Considering the extensive time span following the event, which could potentially influence how stroke and AMI survivors appraise their Quality of Life, we applied the Spearman Correlation test to investigate how the duration since the cardiovascular event relates to the different dimensions of QoL within each survivor group. However, the results indicated no statistically significant differences (*p* > 0.05).

## 4. Discussion

This study aimed to characterize stroke and AMI survivors and compare their perception of QoL. We surveyed 103 stroke survivors and 101 AMI survivors. We found that 44.1% of the patients at the date of the event were under 66 years of age. On the date of the survey response, this percentage was 36.8%. This reveals a significant affectation of individuals of active working age due to vascular disease, who are not yet considered elderly, and who guarantee the development of society. Undoubtedly, the vascular event generates great repercussions for their lives and for the environment in which they are involved [1,26].

It is evident that males represent a larger proportion in both groups, with a more pronounced prevalence observed in the AMI group (around 75%). These findings align with those of other researchers [1,27] who have reported similar data.

The higher propensity to develop vascular disease in the male sex is widely recognized [28]. Scientific knowledge points to, among other factors, the fact that testosterone appears to contribute to increased LDH cholesterol levels. Estrogen, on the other hand, contributes to a reduction in LDH cholesterol and an increase in HDL cholesterol [28,29]. Therefore, according to the same authors, women, until reaching menopause, have a lower risk of developing atherosclerotic disease, based on the hormonal difference in both sexes.

In our sample, 64.8% of the patients had four years or less of schooling and only about 7% had attended university. It is known that the educational level of a population is related to its welfare status. Better educated individuals have a higher occupational status, better housing conditions, and healthier lifestyles [26,30] On the other hand, low educational attainment leads to precarious jobs and wages, together with less information on how to maintain health and less ability to access health services, obtain medication, and follow a treatment plan [30]. 

The multidimensionality of the QoL includes social wellbeing [16]. The involvement of family and friends in the disease process provides the patient with support to get rehabilitated and stay as healthy as possible [1]. In the process of illness, the participation of the family is fundamental, and, in this study, it was found that 35.3% of the survivors had a caregiver, often someone from their own family. Having a caregiver implies that the person is involved in a social support network that is related to positive feelings and advantages for the rehabilitation process and maintenance of a healthy lifestyle. However, the need for a caregiver usually reflects physical or psychological dependence and loss of function, due to age or the sequelae of illness [1,31]. 

The approval of fibrinolysis and coronary angioplasty for the treatment of stroke and AMI [32], respectively, conditioned by the therapeutic time intervals, led to a change in the approach procedures when these vascular events were suspected. Protocols have been developed that aim to shorten the time from patient arrival to differentiated treatment, if indicated, to reduce associated morbidity and mortality and increase the QoL after the event.

In our study, we found a substantial difference between the percentage of activations of the Coronary Green Line and the stroke Green Line, since in the first group, it was activated in 37.6% of the cases, and in the second group, it was activated in 15.5% of the patients. This result is related to the fact that the hospital where these patients were admitted was the only reference center in the whole province for fibrinolysis and primary angioplasty. In addition, the therapeutic window for stroke fibrinolysis is 3 to 4.5 h, and for primary angioplasty, it is 12 h after stroke onset, which has an impact on the activation of the ”Via Verde”. Only about 15% of the surveyed stroke patients arrived at the hospital in the therapeutic window for fibrinolysis.

Depression and anxiety together accounted for 27.9% of patients in our sample. Depression and anxiety tend to decrease the QoL as much or more than physical disability [33]. In patients with hemiparesis, the prevalence of depression in the first year after stroke can be as high as 50%, which may increase over time. This seems to reflect altered brain chemistry as stroke in the left frontal lobe seems to also increase its likelihood in some subcortical areas [34,35]. The recovery of social roles, response to rehabilitation, decreased isolation, treatment of depression, and physical recovery post-stroke, with autonomy in performing daily living activities, can enhance the QoL [36,37]. In our research, in all disease groups, the percentage of patients suffering from depression was higher than in the general population [38]; in the stroke group, this percentage reached 19%. Depression has a strong impact on post-event functional recovery, the QoL, and long-term mortality, so early diagnosis and timely treatment is essential for better physical and emotional rehabilitation and to ensure the best possible QoL [35].

Patients who survive a stroke, in most cases, have sequelae that limit functional independence and the satisfaction of their basic needs, which affects their QoL [1]. The post-stroke QoL depends on the severity of neurological deficits, but is usually low, and although it may improve with the resolution of sequelae, the pre-event level of QoL may never be reached [1,26]. This work supports this assertion since most stroke survivors have sequelae of the event (83.5%). Stroke is one of the main causes of disability in the world [1,26], and post-vascular event rehabilitation programs enable patients to recover their deficits or learn to live with them in such a way that they achieve greater independence in their daily activities and, with that, greater satisfaction and a perception of higher QoL.

The semiology of heart failure is very limiting to the QoL. The NYHA functional classification provides a simple means of classifying the severity of heart failure [25]. In this study, about 60% of the patients did not show signs of heart failure. Some patients reported that their QoL had even increased after AMI. This can be explained by the fact that the patient with ischemic heart disease who, for example, had to stop climbing stairs because of precordial pain before the infarction, in the post-AMI period, when he underwent catheterization with stent placement in all the coronary arteries with significant atheroma, has an optimized blood supply to the myocardium, which improves his ability to carry out his daily activities and, therefore, his perception of his QoL [14].

From the analysis of the data, it can be concluded that AMI survivors exhibit higher mean values than stroke survivors in the perception of their QoL. There were even AMI survivors who, when surveyed, reported that their QoL had increased after the AMI. These are, as mentioned above, patients with exertional angina, who after suffering AMI and undergoing angioplasty were left without sequelae or physical limitation and therefore perceive their QoL to be better today. In addition, most AMI survivors underwent angioplasty, which is the gold standard for AMI treatment, but only about 15% of stroke patients underwent fibrinolysis, the gold standard for stroke treatment.

In relation to the cardiovascular risk profile, it seems that, in this sample, hypertension and a sedentary lifestyle were more associated with stroke and that smoking and obesity were more associated with AMI. Smoking control is the most effective measure of the secondary prevention of AMI, and its use should be vigorously combated by providing the patient with a longer and better Quality of Life [2].

## 5. Conclusions

This study aimed to explore and compare how surviving a vascular event, such as AMI or stroke, affects different aspects of individuals’ lives, specifically focusing on their perception of their QoL.

Of the 204 patients, 103 were stroke survivors, with an average age at the time of the survey of 68.7 years, average age at the event of 66.1 years, and average length of time since the event of 30.65 months. The majority were male, were married or cohabiting, had up to six years of schooling, were retired, had a salary below the national minimum wage, lived in the city with a spouse, and were not in need of a caregiver.

With regard to the clinical variables, we can see that the vast majority of patients were not admitted under the Via Verde protocol, and only two patients had no cardiovascular risk factors at the date of the event, with the most prevalent being hypertension and dyslipidemia, and the most frequent comorbidities were depression and anxiety.

Regarding the comparison of QoL perception between the two groups, findings indicate that patients in the AMI group reported a more favorable perception of their QoL than those in the stroke group, specifically in the physical and psychological dimensions, as well as the general facet. Additionally, when examining the comparative cardiovascular risk factors among the survivor groups, it was revealed that hypertension and a sedentary lifestyle were more closely linked to stroke, while smoking and obesity were more prevalent among those who experienced AMI. Notably, stroke survivors were more inclined towards institutionalization, requiring caregiver assistance, experiencing more aftereffects, and displaying poorer mental health indicators.

While various factors may influence the Quality of Life (QoL) perception among survivors of stroke and AMI, the findings underscore the impact of these vascular events on an individual’s QoL, especially concerning their physical condition, functionality, and independence. Therefore, a strong association emerges between functionality and a more positive QoL perception.

There also remains the certainty of the breadth, multidimensionality, and subjectivity of the concept of QoL in this post-cardiovascular event context. A better perception of QoL is related to the state of health but does not depend solely on it. A patient may be a carrier of a disease and perceive his or her QoL to be better because he or she has adapted and lives with different goals and perspectives than a biologically healthy person. Surviving a cardiovascular event and perhaps a disease means overcoming its outcome and recovering, as far as possible, from its outcomes, changing one’s lifestyle, rebuilding an identity, and repositioning oneself in one’s environment. The data presented here should be considered within certain limitations. While the QoL assessment tool used is widely recognized nationally and internationally, employing a standardized instrument to capture data on such a complex subject might have constrained participants from expressing their feelings more extensively. In the future, enhancing this study could involve a mixed methodological approach, combining various research methods. Additionally, conducting longitudinal follow-ups with individuals who have survived such events would be crucial to gaining deeper insight into how different stages of the rehabilitation process impact their perception of their QoL. Retrospective evaluations, like the one conducted here, come with the awareness that numerous uncontrollable variables might exist, potentially introducing biases into the analysis of results.

It is also important to acknowledge that the chosen sampling method and the established inclusion and exclusion criteria could possess certain limitations. Accessibility sampling may present potential bias and might limit the extent of generalization. Additionally, including individuals with diverse post-event periods underscores potential variations in recovery stages, which could potentially influence the study’s outcomes or interpretations.

## Figures and Tables

**Table 1 healthcare-12-00254-t001:** Sociodemographic variables according to the type of event (n = 204).

		STROKE n = 103 n (%)	AMI n = 101 n (%)	Total	*p* Value
Age at the event		$\bar{x}$ = 67.7	$\bar{x}$ = 64.5	$\bar{x}$ = 66.1 SD = 11.9 (Min. 34–Max. 92)	>0.05
Gender	Male	66 (64.1%)	76 (75.2%)	142 (69.6%)	>0.05
	Female	37 (35.9%)	25 (24.8%)	62 (30.4%)	
Marital status	Single	6 (5.8%)	14 (13.9%)	20 (9.8%)	>0.05
	Married/Unmarried couple	66 (64.1%)	58 (57.4%)	124 (60.8%)	
	Widowed	21 (20.4%)	15 (14.9%)	36 (17.7%)	
	Divorced/Separated	10 (9.7%)	14 (13.9%)	24 (11.7%)	
Formal Education	Uneducated	20 (19.5%)	16 (15.9%)	36 (17.6%)	>0.05
	Up to 6 years	57 (55.4%)	40 (39.6%)	97 (47.5%)	
	Up to 12 years	20 (19.5%)	36 (35.6%)	56 (27.5%)	
	Bachelor/Degree	6 (5.8%)	9 (8.9%)	15 (7.4%)	
Employment status	Employed	17 (16.5%)	39 (38.6%)	56 (27.5%)	0.002
	Unemployed	3 (2.9%)	4 (4%)	7 (3.45)	
	Sick leave	14 (13.6%)	6 (5.95)	20 (9.8%)	
	Retired	69 (67%)	52 (51.5%)	121 (59.3%)	
Family income	Less than minimal wage	47 (45.6%)	41 (40.6%)	88 (43.2%)	>0.05
	Between 1 and 2 minimal wages	43 (41.7%)	44 (43.6%)	87 (42.6%)	
	3 or more minimal wages	13 (12.6%)	16 (15.8%)	29 (14.2%)	
Place of residence	Urban	52 (50.5%)	65 (64.4%)	117 (57.4%)	>0.05
	Rural	51 (49.5%)	36 (35.6%)	87 (42.6%)	
Familiar unit	Alone	12 (11.7%)	26 (25.7%)	38 (18.6%)	<0.001
	Institutionalized	18 (17.5%)	4 (4%)	22 (10.8%)	
	With family/others	73 (70.8%)	71 (70.3%)	144 (70.6%)	
Existence of caregiver?	Yes	51 (49.5%)	21 (20.8%)	72 (35.3%)	<0.001
	No	52 (50.5%)	80 (79.2%)	132 (64.7%)	

**Table 2 healthcare-12-00254-t002:** Clinical variables according to the type of event (n = 204).

		STROKE n = 103 n (%)	AMI n = 101 n (%)	Total	*p* Value
“Vía Verde” Admission	Yes	16 (15.5%)	38 (37.6%)	54 (26.5%)	<0.001
	No	87 (84.5)	63 (62.4%)	150 (73.5%)	
Cardiovascular Risk Factor	Hypertension	85 (82.5%)	69 (68.3%)	154 (75.5%)	0.018
	Dyslipidemia	61 (59.2%)	71 (70.3%)	132 (64.7%)	>0.05
	Sedentary lifestyle	51 (49.5%)	27 (26.7%)	78 (38.2%)	<0.001
	Type 2 Diabetes Mellitus	32 (31.1%)	33 (32.7%)	65 (31.9%)	>0.05
	Atrial fibrillation	24 (23.3%)	11 (10.9%)	35 (17.2%)	0.019
	Alcoholism	22 (21.4%)	16 (15.8%)	38 (18.6%)	>0.05
	Smoking	16 (15.5%)	44 (43.6%)	60 (29.4%)	<0.001
	Obesity	15 (14.6%)	28 (27.7%)	43 (21.1%)	0.021
	Ischemic heart disease	2 (1.9%)	18 (17.8%)	20 (9.8%)	>0.05
	Former smoker	10 (9.7%)	18 (17.8%)	28 (13.7%)	>0.05
Clinical evolution	Positive without sequelae	30 (29.2%)	67 (66.4%)	97 (47.5%)	<0.001
	Negative with sequelae	73 (70.8%)	34 (33.6%)	107 (52.5%)	
Mental Health	Depression	20 (19.4%)	11 (10.9%)	31 (15.19%)	>0.05
	Anxiety	17 (16.5%)	9 (8.9%)	26 (12.75%)	>0.05
mRankin Scale actual	No symptoms	40 (38.8%)	-	-	-
	No significant disability	16 (15.5%	-	-	-
	Slight disability	19 (18.4%)	-	-	-
	Moderate disability	12 (11.7%)	-	-	-
	Moderate severe disability	13 (12.6%)	-	-	-
	Severe disability	3 (2.9%)	-	-	-
NYHA Classification	Class I—No symptoms	-	60 (59.4%)	-	-
	Class II—Mild symptoms	-	31 (30.8%)	-	-
	Class III—Marked limitation	-	9 (8.9%)	-	-
	Class IV—Severe limitations	-	1 (0.9%)	-	-

**Table 3 healthcare-12-00254-t003:** Mean difference in the perception of QoL between stroke and AMI survivors.

	Cardiovascular Event	n	$\bar{x}$	U	*p*
General	Stroke	103	57,864	6289.500	0.009
	AMI	101	64,653		
Physical	Stroke	103	213,495	6842.500	<0.001
	AMI	101	241,881		
Psychological	Stroke	103	196,214	6893.000	<0.001
	AMI	101	221,386		

## Data Availability

Authors agree to make data and materials supporting the results or analyses presented in their paper available upon reasonable request.

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
