# Peer review of "A Comparative Study on the Quality of Life of Survivors of Stroke and Acute Myocardial Infarction"

_healthcare, 2024, doi:10.3390/healthcare12020254_

Round 1

Reviewer 1 Report

Comments and Suggestions for Authors

Eva Lourenco et al present a study on the quality of life of patients after stroke or myocardial infarction in a Portuguese region. A total of 204 patients were interviewed and a difference was found between the two groups, with a poorer quality of life in patients after a stroke. The manuscript is clearly written, but some major improvements are needed in the analysis and interpretation of the data, as many aspects have been omitted.

1. The underlying data was collected in 2018 and is therefore unfortunately already five years old. It is unclear why the data are only being published now. An explanation from the authors should be included here, as there have been significant changes in medical care in the last five years, so it cannot be assumed that the statements are still valid today. 

2. It is not clear how long after the stroke or heart attack the patients were interviewed. It is stated that the stroke should have occurred at least 3 months ago and the heart attack at least 6 months ago. However, it is of course relevant to the research question whether a patient had a stroke four months or two years ago and has learnt to live with the resulting disability. This aspect should be taken into account and perhaps shown in a box plot diagram. 

3. Authors should indicate how many patient records were screened for inclusion and how many patients could not be included in the study due to severe impairment or similar reasons. 

4. The questionnaires used should be made available as a supplement. 

5. In table 1 and 2 the word "table" is written twice, please delete once.

6. Lines 191-199 are a basic description of the Rankin Scale and the NYHA Scale and should  be in the Methods, not the Results section.

7. As "Healthcare" has an international readership, the "Via Verde" should be explained. It is unclear what the treatment pathway is. 

8. In line 205, 2% should be in brackets.

9. In line 215, the authors write that more than 25% of the participants have a depression or anxiety disorder. It is not clear whether these diagnoses developed after the stroke or heart attack or were present before. However, this difference naturally has a direct impact on the results, as having a pre-existing depression or anxiety disorder has a significant impact on quality of life.

10. In lines 317-322 the authors write that some patients feel better after a heart attack than before. Can the questionnaire detect this or is it just speculation?

11. In my opinion, incorrect conclusions are often drawn from the available data. In my opinion, the simple comparison of QoL between the two groups is not appropriate because quality of life depends on many factors and cannot simply be reduced to stroke or heart attack. Subgroup analyses should be done to show, for example, whether patients who are more severely affected by a stroke/heart attack have a worse quality of life than those in the same group who are less affected. Furthermore, you write, for example, in line 273 ff. that patients feel better with a social network. However, it is not analyzed whether the social network had an influence on QoL in this study. This should be done when these statements are included in the discussion.

12. A section on limitations should be included in which the authors critically examine the results of their own study and present potential biases.

13. Approximately 30% of the literature used is not written in English, so it is not possible to check these references.

Reviewer 2 Report

Comments and Suggestions for Authors

The manuscript by Lourenço and colleagues evaluates the sociodemographic and clinical characteristics of stroke and AMI survivors and compares their perception of quality of life. Overall, the manuscript is well-written, and this is an interesting study. I have a few comments that I think could help strengthen the presentation of the methods and results.

  • Some sentences in the result section could go to the method section instead. For example, lines 233-237.
  • Why did you use p<0.01 to determine statistical significance? Using a 2-sided p-value of 0.05 is more conventional to indicate the statistical significance.
  • Could you also conduct statistical tests to assess the differences in the sociodemographic variables between the stroke and the AMI group? The p-values could be included in Table 1. Similarly, could you conduct statistical tests to compare the differences in the clinical variables between the stroke and the AMI group? The p-values could be included in Table 2. These results could help evaluate what characteristics of the AMI group could be associated with their higher perception of QoL compared to the stroke group. Could you also include these in the discussion?
  • What does the 𝒙 indicate in table 3? Why are the values in the 𝒙 column so high if these are the sum of the QoL questions, each on a Likert scale?
  • In line 153, you mentioned obtaining a Cronbach’s alpha from 0.68 to 0.89. Could you provide the relevant data to support this statement?
  • What is the distribution of the perception of QoL (general, physical, psychological) in your study participants? Since they are derived from several questions on Likert scales, I imagine they are not normally distributed. Could you use a non-parametric test for the comparisons in Table 3 instead of a student t-test? For example, the Wilcoxon rank sum test would be a good fit.
  • Were there any missing data for the variables in Tables 1 or 2? Or were there any participants who failed to respond to the QoL questionnaire? If some of the participants did not respond to all the questions, how did you handle missing data?
Comments on the Quality of English Language

The writing is unclear and has some minor grammar errors. The English language should be improved to ensure that an international audience can clearly understand your text. Some examples where the language could be improved:

  • In line 24, please describe what variables instead of just saying “affected by several variables,” Or you could say “affected by several sociodemographic and clinical factors.”
  • In line 163, please change “the survey garnered responses from 204” to “204 responded to the survey.”
  • In line 164, please change “current age averaged was 68.7 years” to “The average age of the study participants was 68.7 years.”
  • In line 170, please change “regarding academic differentiation” to “regarding the education attainment.”

I suggest you have a colleague who is proficient in English and familiar with the subject matter review your manuscript or contact a professional editing service.

Author Response

We would like to thank you very much for your time and willingness to review our article. Your opinion really means a lot to us and is an excellent opportunity for us to reflect on our work and improve its quality. We will endeavour to respond as best we can to your questions and suggestions. Please see the attachment. 

Round 2

Reviewer 1 Report

Comments and Suggestions for Authors

I would like to thank the authors for revising the manuscript, which has improved. However, there are still a few minor points that I would like to address:

1. Perhaps I did not express it correctly in the first review round, but it is not clear for which time period you selected the patients. You now write that the longest past event was 240 months ago (AMI). Does this mean that you searched the Health Service database for potential study participants with an event in the last 20 years? Can you comment on this in the methodology section?

2. If you have searched the database for events in the last 20 years, the potential number of participants of 224 seems very low. 

3. I would not end the manuscript with the limitations, but with the conclusions. Perhaps the two sections could be swapped.

4. Even if your questionnaire is in Portuguese, you could upload it as a supplement and attach a simple English translation. 

5. Line 260, there is a ".," problem after Mann-Whitney U test.

Comments on the Quality of English Language

Line 265: I doubt that "employed" is the right word at this point. 

Reviewer 2 Report

Comments and Suggestions for Authors

The authors have adequately addressed my comments. Therefore, I have no further comments. 

Author Response

We are grateful for your feedback as it presents us with the chance to enhance and refine our work.